# Self-Powered Biosensor for Specifically Detecting Creatinine in Real Time Based on the Piezo-Enzymatic-Reaction Effect of Enzyme-Modified ZnO Nanowires

**DOI:** 10.3390/bios11090342

**Published:** 2021-09-16

**Authors:** Meng Wang, Guangting Zi, Jiajun Liu, Yutong Song, Xishan Zhao, Qi Wang, Tianming Zhao

**Affiliations:** College of Sciences, Northeastern University, Shenyang 110819, China; 2000176@stu.neu.edu.cn (M.W.); 20191534@stu.neu.edu.cn (G.Z.); 2000165@stu.neu.edu.cn (J.L.); 2000172@stu.neu.edu.cn (Y.S.); 2000189@stu.neu.edu.cn (X.Z.)

**Keywords:** creatinine, ZnO nanowires, piezo-enzymatic-reaction effect, self-powered biosensor

## Abstract

Creatinine has become an important indicator for the early detection of uremia. However, due to the disadvantages of external power supply and large volume, some commercial devices for detecting creatinine concentration have lost a lot of popularity in everyday life. This paper describes the development of a self-powered biosensor for detecting creatinine in sweat. The biosensor can detect human creatinine levels in real time without the need for an external power source, providing information about the body’s overall health. The piezoelectric output voltage of creatininase/creatinase/sarcosine oxidase-modified ZnO nanowires (NWs) is significantly dependent on the creatinine concentration due to the coupling effect of the piezoelectric effect and enzymatic reaction (piezo-enzymatic-reaction effect), which can be regarded as both electrical energy and biosensing signal. Our results can be used for the detection of creatinine levels in the human body and have great potential in the prediction of related diseases.

## 1. Introduction

Following the fast development of the social economy, people’s demands for a better life are not just based on food or warm clothes but also on good health [1,2,3]. The maintenance of a healthy lifestyle is inextricably linked to the monitoring of various body indexes [4,5], with creatinine serving as an important index for the early detection of uremia [6]. Creatinine [6,7,8], one of the most important metabolic health indicators [9,10,11], can indicate abnormal renal function as well as other serious diseases [12,13,14]. However, almost all traditional creatinine sensors require external power supplies and large volumes [15,16], making the sensor inconvenient to transport and posing a risk due to the frequent charging/discharging process [17]. Those features restrict the applications of those devices in many areas such that they cannot be used for daily health detection.

To solve the common problem of portability in capacitor/resistance-based sensors [18], many scientists have conducted in-depth research [19]. Using piezoelectric materials or triboelectric structures to realize self-powered sensors has been a popular solution in recent years. The self-powered nanogenerators can convert external mechanical energy into electrical energy [20,21], and the output is easily affected by the external environment [22] by doping noble metal elements or modifying functional materials [23,24,25,26,27,28,29,30,31], which can be viewed as a sensing signal. Traditional triboelectric nanogenerators based on a vertical contraction–separation structure or a contact–slide structure are not flexible enough due to their unique structure [32,33]. Furthermore, common piezoelectric semiconductors are incapable of detecting biological materials [34]. Therefore, it is essential to design a self-powered sensor [35] for specifically detecting creatinine.

In this paper, a self-powered biosensor for specifically detecting creatinine in sweat is presented. Based on the coupling effect of the piezoelectric effect and enzymatic reaction [36,37,38] (piezo-enzymatic-reaction effect) of the creatininase/creatinase/sarcosine oxidase-modified ZnO NWs, the biosensor can actively output electrical signals, which can be seen as biological signals. In addition, the piezoelectric output voltage is significantly dependent on the creatinine concentration, and therefore can reflect the concentration of creatinine. Hence, the whole process does not need an external power supply. Our research results represent a great contribution to the detection of creatinine levels in the human body.

## 2. Materials and Methods

### 2.1. Materials

Zinc acetate (Zn(CH_3_COO)_2_·2H_2_O), zinc nitrate hexahydrate (Zn(NO_3_)_2_·6H_2_O), creatininase, creatinase, and sarcosine oxidase were purchased from Macklin Inc. (Shanghai, China) Ammonia solution (NH_3_·H_2_O), creatinine, and phosphate-buffered saline (PBS) solution were provided from Sinopharm Chemical Reagent Co. Ltd. (Shanghai, China) Ti film (200 μm) and Kapton film (100 μm) were purchased from Taobao (Hangzhou, China). Photoresist and developing solutions were purchased from Suzhou Ruihong Electronic Chemicals Co. Ltd (Suzhou, China). All chemicals were analytically pure and utilized without further purification.

### 2.2. Synthesis of ZnO NWs

The ZnO NWs were synthesized by a simple seed-assisted hydrothermal method. First, a ZnO seed layer was grown on Ti foil. A precleaned Ti film was immersed in a Zn(CH_3_COO)_2_·2H_2_O (10 mM in ethanol) solution for several seconds and then dried under nitrogen gas. The Ti film with Zn(CH_3_COO)_2_ nanoparticles was then annealed at 350 °C for 20 min to form the ZnO seed layer. Second, the ZnO NWs were synthesized on the ZnO seeds. The Ti film with the ZnO seed layer was immersed into a solution containing 0.82 g of (Zn(NO_3_)_2_·6H_2_O) and 2.5 mL of NH_3_·H_2_O in 38 mL of deionized water. After 24 h at 83 °C, the Ti film with the ZnO NWs was cleaned with deionized water and ethanol several times; the Ti foil was recycled for the next use after the transfer-printing process to further lower the fabrication cost.

### 2.3. Device Fabrication

The fabrication process of the device is mainly composed of photolithography, electron-beam evaporation, and culture process. The experimental process is described below in detail. First, the ZnO NWs of the Ti film were transferred to a Kapton film with a knife. Then, a ~2 μm photoresistor was spin-coated on the Kapton film at 2000 rpm for 70 s. Using a lithography process, a predesigned pattern was formed. Next, the Ti electrodes were formed by an electronic beam evaporation process. The thickness of the Ti electrodes was set at 200 nm. After removing the residual photoresist, the device was dried overnight with nitrogen gas. Finally, creatininase (10 u/mL), creatininase (10 u/mL), and sarcosine oxidase (10 u/mL) were added dropwise on the device and an incubation procedure was conducted in a biosafety cabinet for 2 h. For a better test outcome, the creatininase, creatinase, and sarcosine oxidase were dissolved in a PBS solution (pH ~7.4) at 20 °C. The self-powered creatinine biosensor was stored at 4 °C.

### 2.4. Characterization and Measurement

A scanning electron microscope (SEM; JEOL JSM-6700 F) equipped with an energy dispersive X-ray spectrometer (EDS) was used to characterize the devices’ morphology. A low-noise preamplifier was used to measure the performance of the devices (Model SR560, Stanford Research Systems). A programmable system with a stepping motor and a sliding rail provided the force and working frequency applied to the devices. All experiments were carried out at a temperature of 25 °C and relative humidity of 40%.

## 3. Results

In recent years, creatinine has increasingly been tested in daily life, as an important biomarker to measure human health [39,40]. However, the majority of commercial instruments require an external power source, which make them bulky and unwieldy, stifling further development. In this study, a self-powered creatinine biosensor based on enzyme-modified ZnO NWs is demonstrated. Combining the piezoelectric effect of ZnO NWs with enzyme reaction (piezo-enzymatic-reaction effect), the device can actively output electrical pulses by harvesting tiny mechanical energy, which is significantly dependent on creatinine concentration. With increasing creatinine concentration, the piezoelectric output voltage decreases. Furthermore, because the piezoelectric output voltage can be regarded as both a power source and a biosignal, the entire process does not require any power supply.

Figure 1a shows the concept of a self-powered creatinine biosensor and Figure 1b shows an optical photograph of the self-powered creatinine biosensor. The predesigned pattern includes many interdigital electrode pairs. The distance between electrodes is 4 μm and the distance between each electrode pair is 20 μm to prevent electrical short circuit because the length of the ZnO NWs is ~7 μm (greater than 4 μm, less than 20 μm). Figure 1c shows the 45° view SEM of the ZnO NWs. It can be observed that the ZnO NWs are orderly grown on the Ti film and the diameter of the ZnO NWs is ~200 nm. In Figure 1d, a side view SEM of the ZnO NWs shows that the average length is ~7 μm. The length ensures the ZnO NWs cannot cross electrode pairs (less than 20 μm). Figure 1e shows a single ZnO NW bridge and electrode pair before removing the photoresist. The distance between an electrode pair is 4 μm. Figure 1f shows a single ZnO NW bridge and electrode pair after e-beam evaporating the Ti electrodes. It can be seen that the distance between each electrode pair is 20 μm. Figure 1g shows the X-ray diffraction pattern of the ZnO NWs. The triangle represents the characteristic peak of ZnO and the circle represents the characteristic peak of Ti. The peaks around 31.8°, 34.4°, 36.2°, 47.5°, 56.6°, 62.9°, 69.1°, 72.6°, 76.9°, 81.4°, and 89.6° correspond to the (100), (002), (101), (102), (110), (103), (200), (112), (201), (004), and (202) (PDF#89-0511), respectively. In addition, the peaks around 38.4°, 44.6°, and 64.9° can be attributed to the Ti substrate. Figure 1h shows the EDS spectrum of the ZnO NWs. The peaks of the Zn and O elements contribute to the ZnO NWs and the peaks of Ti contribute to the Ti film.

Figure 2 shows the fabrication of the self-powered creatinine biosensor. The process includes synthesis, transfer printing, photolithography, electron-beam evaporation of the Ti electrode, and enzyme culture of the ZnO NWs. First, during the hydrothermal synthesis of the ZnO NWs [36,37,41], the length of the ZnO NWs was adjusted by controlling the pH value of the solution and the synthesis time during the whole process. To ensure that the ZnO NWs can have the same orientation on the Kapton substrate, the vertical ZnO NWs need to be gently scraped with a knife for the transfer process. Then, the creatinine detector was prepared by photolithography, electron-beam evaporation, and enzyme modification on the prepared material. The specific process can be found in the Materials and Methods.

The piezoelectric properties of the device were measured as shown in Figure 3. Figure 3a shows the schematic diagram of the measurement system. The device is put in a Petri dish filled with a PBS solution to mimic the sweat environment. To apply the working force and frequency, a programmable system with a stepping motor and a sliding rail is used. The device’s electrodes are linked to the SR560 preamplifier (Model SR560, Stanford Research Systems), which collects piezoelectric signals and feeds them into a computer to monitor real-time voltage changes. A computer collects the piezoelectric output voltage. Figure 3b depicts the relationship between applied force and piezoelectric output voltage at 1 Hz and 0°, with the applied force shown in the inset. When the applied forces are 19, 22, and 25 N, the piezoelectric output voltages of the device are 0.24, 0.36, and 0.56 V, respectively. With the increase in pressure, the voltage value significantly increases. In addition, the linear relationship between the force and piezoelectric voltage output is:(1)y=−0.91083+0.05933×F,
where *y* represents the piezoelectric output voltage (in V) and *F* represents the applied force (in N). The square of the correlation coefficient is 0.93762, showing a great fit. Figure 3c shows the relationship between the bending angles and piezoelectric output voltage at 22 N and 1 Hz; the bending angles are defined as shown in the inset of Figure 3c. At bending angles of 30°, 45°, and 60°, the piezoelectric output voltages of the device are 0.14 V, 0.31 V, and 0.57 V, respectively. With the increase in bending angles, the piezoelectric output voltage significantly increases. In addition, the linear relationship between the angles and piezoelectric voltage output is:(2)y=−0.33852+0.01523×θ,
where *y* represents the piezoelectric output voltage (in V) and *θ* represents the bending angle (in °). The square of the correlation coefficient is 0.97749, showing a great fit. Figure 3d shows the piezoelectric output voltage against different working frequencies at 22 N. The frequency has little effect on the piezoelectric output voltage. At 2, 1, and 0.5 Hz, the piezoelectric output voltage is almost stable (~0.3 V).

The sensing performances for detecting creatinine concentration are shown in Figure 4. Figure 4a shows the measurement process. To simulate the sweat environment, the self-powered biosensor is placed in a Petri dish filled with a PBS solution, and the solution containing different concentrations of creatinine is added to the Petri dish dropwise. The relationship between creatinine concentration and piezoelectric output voltage is depicted in Figure 4b. When the creatinine concentrations are 1 × 10^−5^, 1 × 10^−4^, 1 × 10^−3^, 1 × 10^−2^, and 1 × 10^−1^ mM, the piezoelectric outputs are 0.65, 0.41, 0.28, 0.21, and 0.12 V, respectively. With the increase in creatinine concentration, the piezoelectric output significantly decreases. Figure 5b shows that the detection range of creatinine concentration of the device is 1 × 10^−5^–1 × 10^−1^ mM. For further analysis, the data between the log of concentration and piezoelectric output voltage is fitted by linear regression. The fit line is:(3)y=0.0885−0.0229×lgC, 
where *y* represents piezoelectric output voltage (in V) and *C* represents creatinine concentration (in mM). The self-powered creatinine sensor has a sensitivity of 0.0229 V/mM. The sensing performance was tested at a concentration of 0.1 mM creatinine against various conditions (applied force, bending angle, and working frequency) shown in Figure 4c–e. Figure 4c shows the influence of creatinine concentration and force on piezoelectric output. When the creatinine concentration is 0 or 1 mM, 19, 22, and 25 N forces are applied to the material. As the creatinine concentration increases and the applied force decreases, the piezoelectric output significantly decreases. Similar results can be observed, which is contributed to the piezo-enzymatic-reaction effect. Figure 4f,g show the comparison of piezoelectric output variation trends under different applied forces, bending angles, and force frequencies before and after adding creatinine. Though the piezoelectric output voltage increases with the increasing applied force and bending angles before adding creatinine, the biosensing output after adding creatinine also increases with the increasing applied force and bending angles, which implies that the biosensing performance has little influence on applied force and bending angles (Figure 4f,g). Similar results can be seen in Figure 4h.

Figure 5 shows the selective specificity of the self-powered creatinine biosensor. The concentration of all solutions used in the experiment is 0.01 mM. Figure 5a shows the piezoelectric output of the device without modifying enzymes. In this experiment, the piezoelectric output voltage of the device without modifying enzymes is similar against saline, glucose, lactic acid, urea, and creatinine. However, as shown in Figure 5b, after adding creatinine (0.001 mM), the piezoelectric output voltage decreases (red line). The addition of other materials (glucose, lactic acid, and carbamide) (0.001 mM) has little effect on the piezoelectric output voltage. Finally, increasing the concentration of the creatinine solution to 0.01 mM lowers the piezoelectric output voltage (yellow line). These findings suggest that enzymatic reactions can influence the piezoelectric output voltage of the ZnO NWs and that creatininase/creatinase/sarcosine oxidase-modified ZnO NWs have the potential to be used in the fabrication of self-powered biosensors. Figure 5c shows the output of the device across creatinine solutions (0.1 mM) for the first four days. It can be seen that piezoelectric output voltage increases on the third day. However, by timely replenishing enzymes, the voltage returns to its initial state.

Table 1 is a comparison of the performance parameters of two common commercial creatinine sensors and the self-powered creatinine biosensor. As shown in the table, the limit of detection parameter of the self-powered creatinine biosensor is lower than that of the two commercial creatinine detectors. Due to its small size and lack of external power supply, the self-powered creatinine biosensor is superior in terms of portability. Therefore, the self-powered creatinine sensor has a good application prospect for daily life.

Figure 6 illustrates the mechanism of the piezo-enzymatic-reaction effect. The simulation results are from COMSOL Multiphysics 5.5. When no force is applied, the piezoelectric output voltage is 0 V, as shown in Figure 6a. When the ZnO NWs are deformed by an applied force, the piezoelectric output voltage is proportional to the c-axis external strain (Figure 6b) [42]. The larger force and higher angles can induce a larger deformation, which improves the output (Figure 3b,c). The enzymatic reaction is shown in Figure 6(c-i) [43]. After the reaction, H_2_O_2_ is produced as:(4)$$Creatinine+H_{2}O+O_{2}\overset{Creatininase/Creatinase/Arcosine\;Oxidase}{\to }\;Formaldehyde+Glycine+H_{2}O_{2},$$

*H*_2_*O*_2_ is unstable and H^+^ and *e^−^* are produced [36] as:(5)$$H_{2}O_{2}\;\to \;2e^{-}+2H^{+}+O_{2}$$

The large number of H^+^ and e^−^ have directional movement and screen the built-in electric field of the ZnO NWs under deformation (Figure 6(c-ii)). The higher concentration of creatinine solution can release more H^+^ and e^−^, which enhance the screen effect and further lower the piezoelectric output voltage. Other materials are weak electrolytes that are difficult to ionize to enhance the screen effect. Therefore, when the enzymatic reaction occurs, the piezoelectric output voltage will decrease.

## 4. Conclusions

In summary, a self-powered creatinine biosensor was built. The device can detect creatinine concentration in sweat in real time using the piezo-enzymatic-reaction effect of ZnO NWs, and the entire sensing process requires no external power supply. The detection range of creatinine concentration of the self-powered creatinine biosensor was 1 × 10^−5^–1 × 10^−1^ mM and the sensitivity was 0.0229 V/mM. This result indicates that the creatinine sensor has a wide application prospect and has great significance for the prevention of related diseases.

## Figures and Tables

**Figure 1 biosensors-11-00342-f001:**
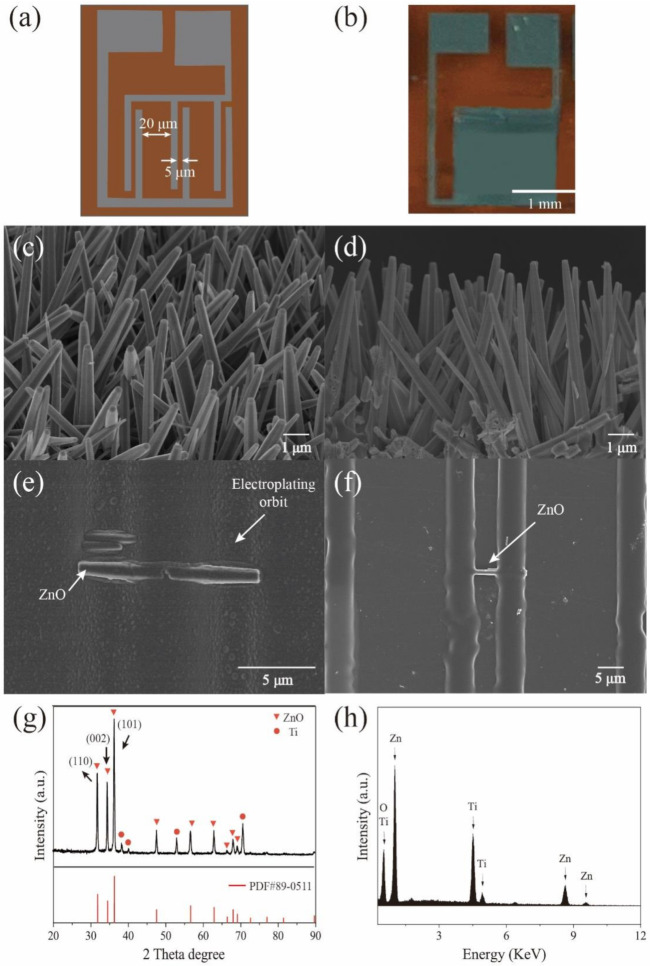
(**a**) Concept of self-powered creatinine biosensor. (**b**) Optical photograph of self-powered creatinine biosensor. (**c**) 45° view SEM of the ZnO NWs. (**d**) Side view SEM of the ZnO NWs. (**e**) A single ZnO NW bridge and electrode pair before removing the photoresist. (**f**) A single ZnO NW bridge and electrode pair after e-beam evaporating the Ti electrodes. (**g**) X-ray diffraction pattern of the ZnO NWs. (**h**) Energy dispersive X-ray spectrum of the ZnO NWs.

**Figure 2 biosensors-11-00342-f002:**
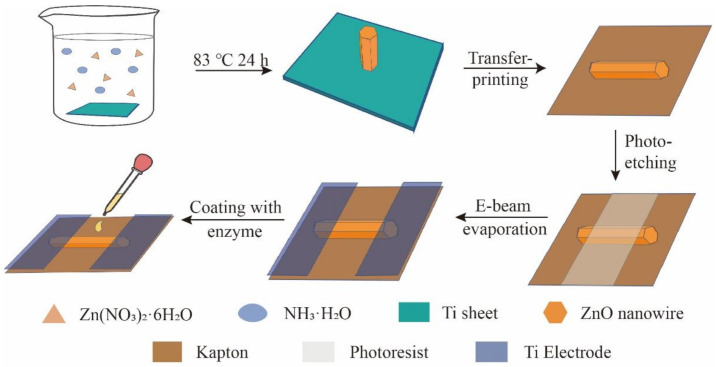
Fabrication of the self-powered creatinine biosensor.

**Figure 3 biosensors-11-00342-f003:**
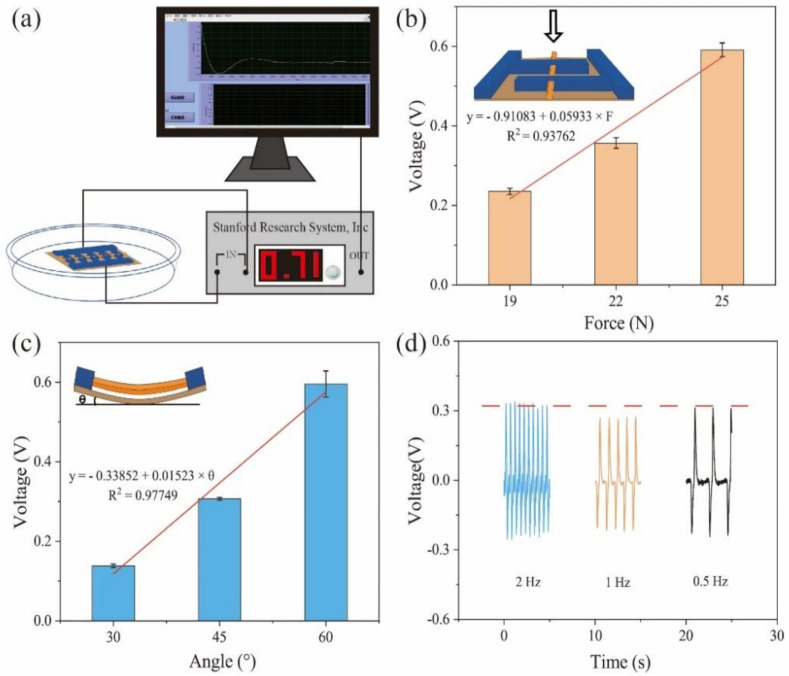
(**a**) Schematic diagram of the measurement system. (**b**) Relationship between applied force and piezoelectric output voltage at 1 Hz; the inset shows the applied force. (**c**) Relationship between the bending angles and piezoelectric output voltage at 22 N and 1 Hz; the bending angles are defined as shown in the inset. (**d**) Piezoelectric output voltage against different working frequencies at 22 N.

**Figure 4 biosensors-11-00342-f004:**
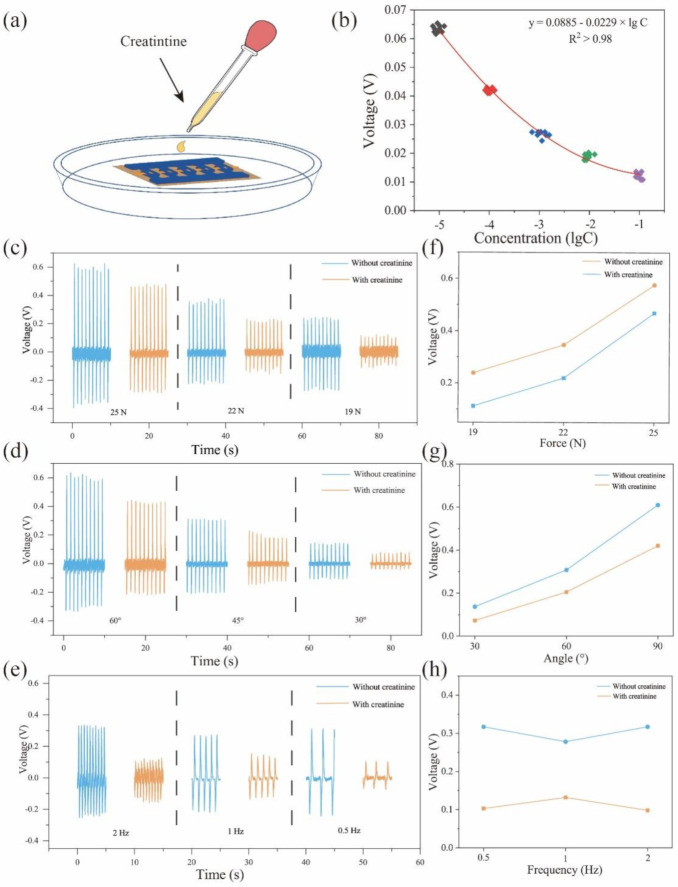
(**a**) Measurement process. (**b**) Relationship between creatinine concentration and piezoelectric output voltage. Sensing performance against different applied forces (**c**), bending angles (**d**), and force frequencies (**e**). Comparison of piezoelectric output variation trends under different force sizes (**f**), bending angle (**g**), and force frequencies (**h**) before and after adding creatinine.

**Figure 5 biosensors-11-00342-f005:**
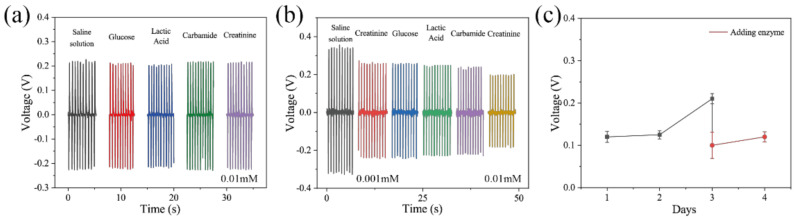
The piezoelectric output voltage of the device before (**a**) and after (**b**) modifying enzymes. (**c**) Output of the device across creatinine solutions for the first four days.

**Figure 6 biosensors-11-00342-f006:**
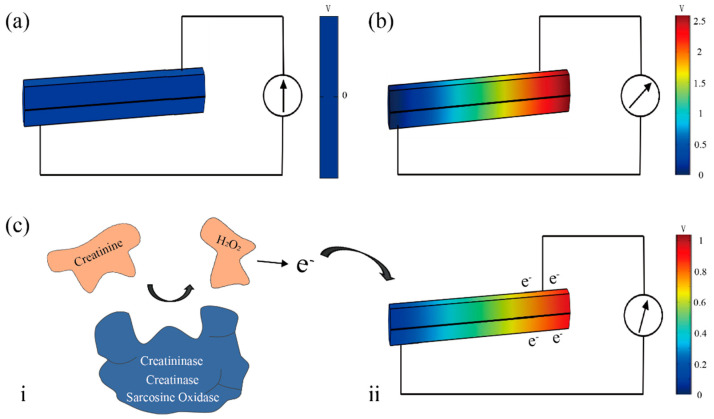
(**a**,**b**) COMSOL simulation of piezoelectric effect. (**c**) Enzymatic reactions and coupling effects.

**Table 1 biosensors-11-00342-t001:** Comparison of performance of three creatinine detectors.

	Limit of Detection	Limit of Quantification	Size	Power Supply	Portability
ACON U120smart	0.08 mM	0.08–2.56 mM	27 × 18 × 14 cm	External power	Not portable
On Call CMU060	0.08 mM	0.08–2.56 mM	14 × 13 × 4 cm	External power	Not portable
This work	1 × 10^−5^ mM	1 × 10^−5^–1 × 10^−1^ mM	2 × 3 × 0.2 mm	Self-powered	Portable

## Data Availability

Data is contained within the article.

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
