# Peer review of "Self-Powered Biosensor for Specifically Detecting Creatinine in Real Time Based on the Piezo-Enzymatic-Reaction Effect of Enzyme-Modified ZnO Nanowires"

_biosensors, 2021, doi:10.3390/bios11090342_

Round 1

Reviewer 1 Report

The authors reported the preparation of a self-powered biosensor for the detection and quantification of creatinine, an important biological indicator of human health. Despite the idea and the general aim of the manuscript are of interest, some important deficiencies and evident errors make the manuscript not acceptable in Biosensors.

In general, all the manuscript is written in a poor English and is quite hard to fluently read it. Some basic mistakes and some typos (some of them severe, such as the pH reported as “PH” in line 111 or the starting a sentence with “And” in line 67), may suggest that the manuscript was written in a hurry (check also “single” in lines 136-139). A deep language revision must be done prior to any other submission.

Last lines of the introduction as well of the abstract claim that the biosensor “will contribute greatly to the prevention of the related diseases”. Authors’ enthusiasm is understandable, but it’s quite ambitious to declare such a future for their device. Maybe (at least) a “can contribute” would be more appropriate, but still some concrete evidence of such a strong sentence must be provided (e.g., scaled up production, real case tests, etc.).

Section 2.2. reports the synthesis of the NW but some amounts are missing (e.g., how much Ti foil was used for each synthesis?). The authors also did not mention that Ti foils are expensive. Will the biosensor still be competitive?

Section 3, lines 103-119, some parts are reported as repetition of section 2.3 without providing any additional information.

XRD analysis in line 131-135 there are no details of the reference for the assignment of the peaks.

The discussion of the results is quite misleading. For example, in line 175-176 is reported that to a voltage of 0.12 V corresponded a concentration of 0.1 mM of creatinine (please note that the molarity is already normalized on the volume (M=mol/L), consequently mM/L doesn’t make any sense) and increasing the creatinine concentration decreased the output voltage. Therefore, one may think that 0.1 mM is the detection limit but in the conclusion section the detection limit is reported to be 0.00001 mM.

Lastly, but still important, no comparison with commonly employed/commercially available creatinine sensors (doeas the new sensor have comparable performances in terms of LOD, LOQ, std.dev. etc.) was carried out in order to demonstrate the real usability of the new device.

Author Response

Thank you for your letter and for the reviewers’ comments concerning our manuscript entitled “Self-powered biosensor for specifically detecting creatinine in real-time based on the piezo-enzymatic-reaction effect of enzyme-modified ZnO nanowire " (Biosensors-1372541). Those comments are all valuable and very helpful for revising and improving our paper, as well as the important guiding significance to our researches. We have studied comments carefully and have made correction which we hope meet with approval. Revised portion are marked in red in the paper. The main corrections in the paper and the responds to the reviewer’s comments are as following attachment.

Reviewer 2 Report

Congratulations on your very good and interesting work, I have just a few simple notes:

1. Who does the sensing the ZnO nanowires or the pieoelectric sensor, this was not very clear.

2. In figure 4, the authors present a series of curves for different forces, who causes this force? And what is the relationship between strength and creatine concentration?

3. It is interesting to present a statistical relationship as the difference between situations with and without creatine are very close.

4. It is necessary to make some minor corrections in English.

Author Response

(The authors gave the same response as above.)

Reviewer 3 Report

A self-powered creatinine biosensor is presented. The article is well organized an experiments done with high scientific standard. The reviewer would recommend to improve figure 1A to help the reader visualize the device easier, add dimensions and perhaps combine with the SEM images of figure 2. From the materials and methods, the reviewer understands that the enzyme is immobilized just by dropcasting without the addition of any membrane (as Nafion) or cross linker (e.g. Gluteraldehyde), as is commonly used. If this is the case, how is the repeatability of the sensor, is it possible to use more than once? How stable is the measurement over time? No optimization for the temperature is shown nor pH, considering that immobilized enzymes are very sensitive to these parameters these should be included. Please indicate the creatinine concentration used along the test shown in Figure 4 (C-E), 5 (A, B). Finally, please indicate explicitly the limit of detection and sensitivity of the obtained sensor.

Author Response

(The authors gave the same response as above.)

Round 2

Reviewer 1 Report

The authors addressed all the issue and remarkably modified the language of the manuscript. The text is now readable, and the presentation of the data sounds much more interesting. Overall the authors did a good job in improving the quality of the paper. I think that the manuscript is now suitable for publication.

Reviewer 3 Report

All concerns addressed by the reviewer were corrected and changes applied to the manuscript. The reviewer therefore positively proposes the current version of the article to be published.